Arsenic disulfide promoted the demethylation of PTPL1 in diffuse large B cell lymphoma cells

Chen Chen 1
Wang Ling 1
Liu Yan 2
Du Shenghong 1
Teng Qingliang tatql@163.com 1
1 Department of Hematology, The Affiliated Taian City Central Hospital of Qingdao University , Taian , Shandong , China
2 Department of Breast Surgery, The Affiliated Taian City Central Hospital of Qingdao University , Taian , Shandong , China
Singh Himanshu
Electronic publication date: 2024 May 14
Publication date: 2024
Volume: 12
Electronic Location ID: e17363
Received 2023 Apr 4; Accepted 2024 Apr 18
Copyright: ©2024 Chen et al.
Copyright year: 2024
Copyright holder: Chen et al.
License: This is an open access article distributed under the terms of the Creative Commons Attribution License, which permits unrestricted use, distribution, reproduction and adaptation in any medium and for any purpose provided that it is properly attributed. For attribution, the original author(s), title, publication source (PeerJ) and either DOI or URL of the article must be cited.
License URL: https://creativecommons.org/licenses/by/4.0/

Keywords: Arsenic disulfide, Diffuse large B cell lymphoma, Demethylation, DNMTs, MBD2

Funding: Taian Science and Technology Development Guiding Project No. 2019NS139 No. 2018NS0172 This work was supported by the Taian Science and Technology Development Guiding Project (No. 2019NS139, No.2018NS0172). The funders had no role in study design, data collection and analysis, decision to publish, or preparation of the manuscript.

==============================
Background

Promoter hypermethylation of the tumor suppressor gene is one of the well-studied causes of cancer development. The drugs that reverse the process by driving demethylation could be a candidate for anticancer therapy. This study was designed to investigate the effects of arsenic disulfide on PTPL1 methylation in diffuse large B cell lymphoma (DLBCL).

Methods

We knocked down the expression of PTPL1 in two DLBCL cell lines (i.e., DB and SU-DHL-4 cells) using siRNA. Then the DLBCL proliferation was determined in the presence of PTPL1 knockdown. The methylation of PTPL1 in DLBCL cells was analyzed by methylation specific PCR (MSPCR). The effect of arsenic disulfide on the PTPL1 methylation was determined in DLBCL cell lines in the presence of different concentrations of arsenic disulfide (5 µM, 10 µM and 20 µM), respectively. To investigate the potential mechanism on the arsenic disulfide-mediated methylation, the mRNA expression of DNMT1, DNMT3B and MBD2 was determined.

Results

PTPL1 functioned as a tumor suppressor gene in DLBCL cells, which was featured by the fact that PTPL1 knockdown promoted the proliferation of DLBCL cells. PTPL1 was found hypermethylated in DLBCL cells. Arsenic disulfide promoted the PTPL1 demethylation in a dose-dependent manner, which was related to the inhibition of DNMTs and the increase of MBD2.

Conclusion

Experimental evidence shows that PTPL1 functions as a tumor suppressor gene in DLBCL progression. PTPL1 hyper-methylation could be reversed by arsenic disulfide in a dose-dependent manner.

Introduction

Diffuse large B cell lymphoma (DLBCL) is the most common subtype of non-Hodgkin’s lymphoma in adults, accounting for about 30–40% (Goldfinger & Cooper, 2022). It is an aggressive malignant lymphoma with untreated median survival of less than 12 months. Patients can achieve a progression-free survival rate of approximately 40% and a long-term survival rate of 50% after the combination of four drugs including doxorubicin, prednisone, vincristine, and cyclophosphamide (Fisher et al., 1993). In addition, rituximab serving as a chimeric anti-CD20 monoclonal antibody further improves the prognosis of DLBCL. Unfortunately, about 20–40% of patients still show no response or even rapid progression after treatment (Chapuy et al., 2018).

Most DLBCL patients have epigenetic heterogeneity, and varied degrees of heterogeneity are related to different prognosis (Cerhan et al., 2014; Jiang & Melnick, 2015). As one form of epigenetic modification, DNA hypermethylation could lead to transcriptional silencing in tumor suppressor genes, thereby leading to the functional loss or attenuation (Lopez et al., 2022). PTPL1 is a protein tyrosine phosphate (PTP) encoded by the human PTPN13 gene, which can exert tumor suppressor role by antagonizing protein tyrosine kinase (PTK) (Freiss & Chalbos, 2011). PTPN13/PTPL1 promoter methylation has been confirmed in a variety of malignancies including non-small cell lung cancer, ovarian cancer, prostate cancer, and breast cancer (Bompard et al., 2002; Castilla et al., 2012; Wang et al., 2022; Wang et al., 2018). In DLBCL and follicular lymphoma, hypermethylation could also be detected in the majority of the PTPL1 gene promoter, along with attenuation or silencing of PTPL1 (Wang et al., 2016). As the epigenetic inheritance of methylation is reversible, activation of tumor suppressor genes induced by pharmacologic demethylation is considered an attractive therapeutic strategy to block tumor growth and progression.

Arsenic trioxide (ATO, As2O3) has been utilized as an anti-cancer agent by suppressing cancers of the liver, prostate, and breast apparently through modulating the DNA demethylation and cellular apoptosis (Thomas et al., 2010; Xia et al., 2012). Arsenic disulfide (As2S2), the main component of traditional Chinese medicine (TCM) realgar, has been reported to show similar antitumor effects and lower toxicity compared with ATO (Skoczynska & Skoczynska, 2022). Zhao et al. (2018) showed that AS2S2 could exert anticancer effects via apoptosis, cell cycle arrest, and pro-survival signal inhibition in human breast cancer cells. In addition, As2S2 induced apoptosis and autophagy through the activation of ROS/JNK and suppression of Akt/mTOR signaling pathways in osteosarcoma (Wang et al., 2017). Furthermore, AS2S2 had obvious antitumor effect on mouse model of human lymphoma transplanted tumor in a dose dependent manner (Wang, Li & Li, 2021).

This study was designed to investigate the effects of arsenic disulfide on PTPL1 methylation in DLBCL. We first investigated the roles of PTPL1 in the proliferation of DLBCL cells by knocking down PTPL1 in two DLBCL cell lines (i.e., DB and SU-DHL-4 cells) using small interfering RNA (siRNA). Then we determined the promoter methylation of PTPL1 in DB and SU-DHL-4 cells in the presence of arsenic disulfide by methylation specific polymerase chain reaction (MSPCR). Finally, we investigated the mechanisms on how arsenic disulfide modulated the PTPL1 methylation by evaluating the mRNA expression of DNMTs (1 and 3B in this case) and methyl-CpG-binding domain 2 (MBD2).

Materials & Methods

Cell culture

DB and SU-DHL-4 cell lines, generously donated by Qilu Hospital of Shandong University, and normal human GM12878 cell line were cultured in Iscove’s modified Dulbecco’s medium (IMDM; Gibco-Invitrogen, Carlsbad, CA, USA) supplemented with 10% fetal bovine serum (FBS; Hyclone, Logan, UT, USA) at 37 °C in an incubator with 5% CO2. Cells in the logarithmic growth phase were collected for subsequent experiments.

Screening of siRNA

To screen siRNA with better knockdown efficiency, DB and SU-DHL-4 cells of logarithmic phase were plated into 6-well plates and transfected using three siRNAs against PTPL1 for 48 h with riboFect™ CP (RIB Bio, Guangzhou, China) according to the manufacturer’s instructions, respectively. Three siRNAs were siRNA1 (sequence, 5′-GGATGATGTTAGTCTAATA-3′), siRNA2 (sequence, 5′-CCACCATGCTGCAATTGAA-3′), and siRNA3 (5′-GCATGAGACTACAAAGACA-3′). Cells transfected with scrambled sequence were used as a negative control (NC) group. The knockdown efficiencies of three siRNAs was verified using reverse transcription PCR (RT-PCR) and Western blotting.

RT-PCR

Total RNA was extracted from cells using TRIzol reagent (Thermo Fisher Scientific, Waltham, MA, USA) according to the manufacturer’s instructions. After the synthesis of cDNA, reverse transcription amplification of PTPL1 was conducted with the specific primers (5′-CAACAATGGTCAGCAACAG-3′; 5′-CACCACAAAGCCCTTCA-3′) that were designed based on the sequence of the transcript CDS regions of the PTPL1 gene searched in NCBI. The amplification conditions were as follows: 95 °C for 5 min, followed by 40 cycles of 95 °C for 10 s and 60 °C for 30 s. GAPDH was used as an internal reference. A fluorescence quantitative PCR instrument (CFX Connect; Bio-Rad, Hercules, CA, USA) was used to analyze the data.

Western blotting

Total protein was extracted from cells in each group using RIPA lysis buffer (P0013D; Beyotime, China). The protein content was evaluated using Pierce™ Rapid Gold BCA Protein Assay Kit (A53225; Thermo Fisher Scientific). Next, the protein was separated on 10% SDS-PAGE gel, transferred onto PVDF membranes (IPVH00010; Millipore, Burlington, MA, USA), and blocked with 5% skimmed milk under room temperature for 1 h. Subsequently, the membranes were incubated overnight at 4 °C with anti-PTPL1 goat polyclonal primary antibody (1:1000; AF3577; R&D) and anti-GAPDH mouse monoclonal primary antibody (1:10000; ab8245; Abcam, Waltham, MA, USA). Then the membranes were washed in Tris-Buffered Saline and Tween (TBST) and incubated with HRP-conjugated donkey anti-goat secondary antibody (1:3000; E-AB-1050; Elabscience, Houston, TX, USA) or HRP-conjugated goat anti-mouse secondary antibody (1:3000; E-AB-1001; Elabscience) at room temperature for 1 h. After washing three times in TBST, images were acquired with an Electro-Chemi-Luminescence (ECL) chemiluminescence kit (P0018M; Biyuntian Bio, Shanghai, China). Signal intensities of bands were quantified using Image J software.

Silencing of PTPL1

To investigate the roles of PTPL1 in DLBCL, DB and SU-DHL-4 cells were divided into the following groups: (i) control group; (ii) negative control (NC) group, transfected with scrambled sequence; (iii) siRNA group, transfected with the selected siRNA from the three siRNAs. Transfection was performed using riboFect™ CP (RIB Bio, Guangzhou, China), according to the manufacturer’s instructions. Briefly, 2 mL of cell suspension was seeded into 6-well plate, and the cell density was adjusted to 1 × 105–5 × 105 cells/well to achieve a cell density of 30–50% at transfection. Cells were then cultured at 37 °C overnight in an incubator with 5% CO2. Next, 10 µL of siRNA solution (20 µM, diluted with 120 µL of 1X riboFect™ CP buffer) and 12 µL riboFect™ CP reagent were added for transfection. The mixture was incubated at room temperature for 15 min and was transferred to a serum-containing medium. After mixing gently, the cells were cultured at 37 °C for 48 h in an incubator with 5% CO2. Cell proliferation was observed using microscopy.

CCK-8 analysis for cellular proliferation

DB and SU-DHL-4 cells (1  × 105/mL, 100 µL) in each group were added into the wells of 96-well plate and incubated in an incubator with 5% CO2 at 37 °C for 48 h. Afterwards, cells were incubated with 10 µL/well CCK-8 reagents for 1 h. Absorbance was measured with a microplate reader at a wavelength of 450 nm. The cell proliferation rate was calculated as the ratio of the OD value in treatment group and the control group.

DNA extraction and methylation

DB and SU-DHL-4 cells in the logarithmic growth phase without any treatment were washed twice with PBS, followed by genomic DNA extraction using a commercial kit (Omega, Norcross, GA, USA), accordance to the manufacturer’s instructions. Afterwards, DNA methylation was performed using 200 ng genomic DNA with a commercial kit (Epigentek, Farmingdale, NY, USA).

MSPCR

MSPCR was conducted to measure the PTPL1 methylation as previously described (Wang et al., 2016). Briefly, two pairs of primers (primer M: 5′-TATAGAAATAAGGTTGAGAGGT AGC-3′, 5′-CGAACGACAAAATTCCTAACG-3′; primer U: 5′-AATATAGAAATAAGGT TGAGAGGTAGT-3′; 5′-ACCAAACAACAAAATTCCTAACAC-3′) were used to amplify methylated DNA and non-methylated DNA, respectively. The amplification conditions were as follows: 95 °C for 5 min, followed by 40 cycles of 95 °C for 30 s, 58 °C for 30 s (for methylated DNA) or 60 °C for 30 s (for non-methylated DNA), and 72 °C for 30 s, and 72 °C for 10 min. Finally, the amplified PCR products were subjected to DNA agarose gel electrophoresis (1.5%), followed by observation using a gel imaging system.

Effects of arsenic disulfide treatment on PTPL1 methylation

To evaluate the effects of As2S2 on PTPL1 methylation, DB, SU-DHL-4 and GM12878 cell lines were treated with 5 µM, 10 µM, and 20 µM of As2S2 (Sigma-Aldrich, St. Louis, MO, USA; dissolved in 1 M NaOH and adjusted pH value to 7.35–7.45 using HCL) for 72 h, respectively. The cells received no arsenic disulfide treatment served as control. Then MSPCR was performed to evaluate the PTPL1 methylation in the presence of arsenic disulfide.

Effects of arsenic disulfide on DNMT1, DNMT3B, and MDB2

To evaluate the effects of arsenic disulfide on given DNMT1, DNMT3B, and MBD2 that in general affect the methylation of the PTPL1 gene, GM12878, DB and SU-DHL-4 cell lines were treated with 5 µM, 10 µM, and 20 µM of arsenic disulfide for 72 h, respectively. The cells without arsenic disulfide treatment were used as control. The mRNA levels of DNMT1, DNMT3B and MBD2 were detected. DNMT1 primers were 5′-CAACGGGCAGATGTTTCA-3′and 5′-TCCTCACATTCATCCACCA-3′. DNMT3B primers were 5′-GAGAAAGCTAGGGTGCGA-3′and 5′-CACTGGTTGCGTGTTGTT-3′. MBD2 primers were 5′-AGTAAGCCCCAGTTGACACG-3′and 5′-AACTGACACAGGCTG CTTGA-3′. GAPDH (5′-ACAACTTTGGTATCGTGGAAGG-3′ and 5′-GCCATCACGCCA CAGTTTC-3′) was used as an internal reference.

Statistical analysis

SPSS 21.0 was used to statistical analysis. One-way analysis of variance (ANOVA) was used for significance test. The significance level was set at P < 0.05.

Results

Selection of optimal siRNAs for the knockdown of PTPL1 gene

Among three siRNAs, siRNA2 (sequence: 5′-CCACCATGCTGCAATTGAA-3′) had the best knockdown efficiency (Fig. 1). Therefore, siRNA2 was used to transfect DB and SU-DHL-4 cells to knock down PTPL1 in subsequent experiments.

Figure 1 Knockdown efficiencies of three siRNAs verified using RT-PCR and Western blotting.

(A) PTPL1 mRNA expression in control, negative control (NC), siRNA1, siRNA2, and siRNA3 groups detected by RT-PCR; (B) Western blotting bands and PTPL1 protein expression in control, NC, siRNA1, siRNA2, and siRNA3 groups. *P < 0.05, **P < 0.01 and ***P < 0.001.

PTPL1 knockdown promoted DLBCL cell proliferation

Micrographs and CCK-8 result together showed that compared with control and NC groups, siRNA group showed significant increase in cellular proliferation rate (P < 0.01, Fig. 2). This indicated that PTPL1 knockdown promoted the proliferation of DB and SU-DHL-4 cells. Therefore, PTPL1 gene served as a tumor suppressor gene in the DLBCL.

Figure 2 Cell proliferation and proliferation rate of control group, NC group and siRNA group in DB and SU-DHL-4 cell lines.

**P < 0.01.

Increased PTPL1 promoter methylation in DLBCL cell lines

Methylation was characterized by the appearance of amplification products of primer M. Unmethylation was characterized by the appearance of amplification products of primer U. The amplification products of both primer M and primer U indicated partial methylation. The amplification of primer M was observed in DB and SU-DHL-4 cell lines (Fig. 3), indicating the methylation in PTPL1 promoter.

Figure 3 PTPL1 methylation in DB and SU-DHL-4 cell lines detected by MSPCR.

Arsenic disulfide inhibited the methylation of PTPL1 in a dose-dependent manner

As2S2 treatment showed no effects on PTPL1 methylation levels in GM12878 cell line, which demonstrated that PTPL1 methylation was not present in normal cell lines (Fig. 4). In contrast, As2S2 treatment significantly reduced PTPL1 methylation levels in both DB and SU-DHL-4 cell lines (P < 0.01). In addition, compared with NC group, the PTPL1 methylation was attenuated in three As2S2 treatment groups (Fig. 5), especially the 20 µM AS2S2 group. These indicated that As2S2 exhibited demethylation role, showing a dose-dependent manner.

Figure 4 PTPL1 methylation levels before and after arsenic disulfide (20 µM) treatment in GM12878, DB and SU-DHL-4 cell lines.

**P < 0.01 and ***P < 0.001.

Figure 5 PTPL1 methylation in DB and SU-DHL-4 cells treated with different doses of arsenic disulfide detected by MSPCR.

Arsenic disulfide regulated DNMT1, DNMT3B and MBD2 mRNA expression

To investigate the potential mechanisms on arsenic disulfide mediated PTPL1 methylation inhibition, we analyzed the mRNA expression of three crucial enzymes involved in the methylation including DNMT1, DNMT3B and MBD2. RT-PCR results showed that arsenic disulfide significantly decreased the mRNA expression of DNMT1 and DNMT3B and significantly increased the mRNA expression of MBD2 (Fig. 6). Such phenomenon showed a dose-dependent manner.

Figure 6 DNMTl, DNMT3B, and MBD2 mRNA expression in DB and SU-DHL-4 cells treated with different doses of arsenic disulfide detected by RT-PCR.

*P < 0.05 and **P < 0.01.

Discussion

DLBCL is the most common lymphoid neoplasm with dismal outcomes (Campo et al., 2011). Greater epigenetic heterogeneity is linked with poor outcome among DCBCL patients (Jiang & Melnick, 2015). Fortunately, the epigenetic alterations provide a number of additional targets that can be pharmacologically modified, which holds the promise for improved outcomes (Shaknovich et al., 2010). PTPL1/PTPN13, encoding a high-molecular-weight (270 kDa) non-receptor type phosphatase (Yeh et al., 2006), is reported to show genetic polymorphisms, and some mutations could lead to deletion of entire catalytic phosphatase domain or inhibition of the phosphatase activity (Zhu et al., 2008). In addition to genetic polymorphisms, epigenetic regulation of PTPN13/PTPL1 expression has been demonstrated in cancers. Therefore, we hypothesized that the regulation of PTPL1 could modulate DLBCL progression. Our results indicated that PTPL1 functioned as a tumor suppressor gene and was found hypermethylated in DLBCL cells. In addition, the arsenic disulfide promoted the demethylation of PTPL1 promoter.

To date, the roles of PTPL1 in the pathogenesis and progression of tumors remains controversial. Li et al. (2000) demonstrated that PTPL1 over-expression increased resistance to Fas-induced apoptosis by the anti-Fas antibody CH-11 in Jurkat and TMK-1 cells. Ungefroren et al. (2001) found that PTPL1 might serve as a potential inhibitor of Fas-mediated apoptosis in pancreatic cancer cells. In astrocytoma cells, the knockdown of PTPL1 led to increased apoptosis and increased sensitivity to Fas-induced cell death (Foehr et al., 2005). Some studies have confirmed that PTPL1 could regulate Fas-mediated apoptosis in colon cancer, melanoma and myeloid cells using SLV inhibitory peptide (Huang et al., 2008; Yao et al., 2004), or RNA interference (Schickel et al., 2010; Xiao et al., 2010). Conversely, PTPL1 expression was sufficient to block the IRS-1/PI3K/Akt signaling pathway to inhibit the insulin-like growth factor-I effect on cell survival and to induce apoptosis (Dromard et al., 2007). In addition, Wang et al. (2022) found that PTPL1 played a crucial suppressive role in the pathogenesis of lung cancer through counteracting the Src/ERK/YAP1 pathway. The findings of Zhu et al. (2021) indicated that the knockdown of PTPL1 enhanced the migration and invasion capabilities of A549 cells through enhancing TGF-β1-induced epithelial–mesenchymal transition. In the present study, we used siRNA to knock down PTPL1 in DB and SU-DHL-4 cell lines to clarify the effects of PTPL1 on DLBCL progression. The results showed that PTPL1 knockdown promoted DLBCL cellular proliferation, which indicated that PTPL1 served as a tumor suppressing gene in DLBCL.

The loss of tumor suppressing gene function can result in malignant growth, usually including self-sufficiency in growth signals, insensitivity to growth-inhibitory signals, evasion of apoptosis, unlimited replicative potential, sustained angiogenesis, and tissue invasion and metastasis (Wang & Wang, 2013). Tumor suppressing gene may be epigenetically silenced by hypermethylation of CpG islands located in their promoter regions (Mehta et al., 2015). PTPL1 was downregulated or silenced in multiple cell lines such as non-Hodgkin lymphoma. Interestingly, methylation of PTPL1 was detected by MSPCR in almost all cell lines with reduced or silenced PTPL1 expression (Yeh et al., 2006; Ying et al., 2006). In this study, we detected the PTPL1 promoter methylation in two DLBCL cell lines (i.e., DB and SUDHL4 cells) using MSPCR. Consistently, our data suggested that PTPL1 was methylated in both cell lines. Interestingly, unlike genetic alterations, DNA methylation is reversible (Kedhari Sundaram et al., 2019). Therefore, demethylation may restore the expression of tumor suppressor genes, and then inhibit tumor progression, which provides a new idea for the treatment of tumors. Recently, some epigenetic drugs (e.g., decitabine and azacitidine) have been approved for the treatment of hematological malignancies (Blecua, Martinez-Verbo & Esteller, 2020). Nevertheless, numerous patients showed recurrence due to poor responses to these drugs (Bazinet & Bravo, 2022). Therefore, there is still a need to develop new drugs targeting DNA methylation.

Our data showed that arsenic disulfide promoted the demethylation of PTPL1 gene. As important TCM components, arsenic drugs, including arsenic disulfide (As2S2), arsenic tetrasulfide (As4S4), ATO (As2O3), exhibit favorable anti-tumor effects in various tumors especially blood-related malignancies (Wang et al., 2013). In 2000, the FDA approved ATO for treating acute promyelocytic leukemia (Jing et al., 1999). Compared with ATO, arsenic disulfide showed comparable anti-tumor effects and more advantages, including lower toxicity of oral administration (Zhao et al., 2019). To date, the mechanism of arsenic disulfide against tumors is still unclear. In a previous study, arsenic disulfide exerted anti-tumor role by induction of autophagy and apoptosis, as well as cell cycle arrest (Zhao et al., 2018). In this study, arsenic disulfide induced decrease of PTPL1 gene methylation in a dose dependent manner. Overexpression of DNMTs (e.g., DNMT1, DNMT3A and DNMT3B) could promote DNA hypermethylation, which was closely related to the prognosis in cancer patients (Weisenberger, Lakshminarasimhan & Liang, 2022). In addition, MBD2 functions as a component of the MeCP1 complex has been reported to act as an epigenetic modulator in various malignancies (Feng & Zhang, 2001; Pei et al., 2019). For the expression of DNMTs, the RNA expression of DNMT1 and DNMT3B showed significant decrease after arsenic disulfide treatment, while the mRNA expression of MBD2 showed significant increase in these cells. These suggested that the inhibition of DNMTs and the increase of MBD2 were potential mechanisms of arsenic disulfide-induced PTPL1 demethylation.

This study has some limitations. First, the optimal dose for arsenic disulfide demethylation remains unclear. Secondly, although we confirmed the demethylation role of arsenic disulfide on methylated PTPL1, the exact mechanisms are still not well defined. Furthermore, biological system does not work in isolation. Although arsenic disulfide treatment induced the demethylation of PTPL1 by reducing the expressions of DNMT1 and DNMT3B, whether arsenic disulfide could cause the disrupted normal methylation pattern remains unknown. More studies in the future are required to focus on the delivery of arsenic disulfide to the PTPL1 DNA promoter using tools such as CRISPR, thereby inducing specific demethylation of PTPL1.

Conclusion

In summary, PTPL1 was a tumor suppressor gene in DLBCL progression. PTPL1 methylation could be reversed by arsenic disulfide in a dose-dependent manner.

Supplemental Information

Supplemental Information 1 Figure 1 raw data.

Supplemental Information 2 Figure 2 raw data.

Supplemental Information 3 Figures 3 and 5 raw data.

Supplemental Information 4 Figure 4 raw data.

Supplemental Information 5 Figure 6 raw data.

Additional Information and Declarations

Competing Interests

Author Contributions

Data Deposition

The authors declare there are no competing interests.

Chen Chen analyzed the data, authored or reviewed drafts of the article, and approved the final draft.

Ling Wang analyzed the data, authored or reviewed drafts of the article, and approved the final draft.

Yan Liu performed the experiments, prepared figures and/or tables, and approved the final draft.

Shenghong Du performed the experiments, prepared figures and/or tables, and approved the final draft.

Qingliang Teng conceived and designed the experiments, authored or reviewed drafts of the article, and approved the final draft.

The following information was supplied regarding data availability:

The raw measurements are available in the Supplementary Files.

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
