# Peer review of "Arsenic disulfide promoted the demethylation of PTPL1 in diffuse large B cell lymphoma cells"

_PeerJ, doi:10.7717/peerj.17363_

## Round 0.1 · original submission · Major Revisions

The reviewers have provided valuable feedback on the manuscript, highlighting areas that require significant revisions. The following is an expanded explanation of the necessary changes:

Introduction:
The introduction should be rewritten to provide better context and background information. It is essential to establish a clear connection between the different parts of the introduction. Additionally, more details should be provided about DLBCL patients, including factors such as gender and survival rates. This will help the readers understand the pathogenesis of DLBCL and the relevance of the study.

Methods:
The methods section needs clarification on specific procedures and should include relevant data for comparison. For instance, in the Materials and Methods section, the description of Reverse Transcription PCR (RT-PCR) should be made more specific. It should be clarified that RT-PCR is used for making cDNA, followed by regular DNA amplification PCR and DNA agarose gel electrophoresis (1.5%). This will ensure that readers have a clear understanding of the techniques used in the study.

Discussion:
The discussion section should address the controversy surrounding PTPL1 as a tumor suppressor or promoter in greater detail. While the manuscript mentions that numerous studies have shown the abnormal expression of PTPL1 in various malignant tumors, it is important to explain how this study contributes to clarifying the controversy. Discuss the findings in the context of previous research and present a comprehensive analysis of the potential role of PTPL1 as a tumor suppressor or promoter in DLBCL. Consider the broader implications of disturbed methylation patterns, both on specific genes and on the overall biological system.

Additional Improvements:
Implement the suggested improvements made by the reviewers, such as adding western blot data if available and providing marker sizes in Figures 2 and 3. These changes will enhance the clarity and completeness of the manuscript.

In summary, the manuscript requires significant revisions to address the reviewers' comments. By rewriting the introduction, clarifying the methods section, and expanding the discussion, the manuscript will provide a more comprehensive and coherent presentation of the research. Additionally, implementing the suggested improvements will improve the overall quality of the manuscript.

Reviewer 1 ·

Basic reporting

Need to rewrite the introduction because most of the part is not interconnecting.

Experimental design

1. Please explain the diffuse large B cell lymphoma (DLBCL) in brief.
2. Need to explain cell culture and its methodology in detail to make readers clear
3. 78-79, Give the brief explanation of specific primers
4. Give the detailed protocol for the Silencing of PTPL1

Validity of the findings

1. Line 125, How PTPL1 knockdown promoted DLBCL cell proliferation and which type
of cell signalling cascade pathway use.
2. 148-149 line please explains.( Its dephosphorylation can dephosphorylate the tyrosine of tyrosine kinase, thereby antagonizing the growth-promoting effect of tyrosine kinase).
3. 150-152, numerous studies have shown the abnormal expression of PTPL1 in
various malignant tumors. However, whether PTPL1 is a tumor suppressor or a
tumor promoter is still controversial. How does this study clarify this controversy? Please explain it in details in discussion part.

Additional comments

NA

·

Basic reporting

a. Clear, unambiguous, professional English language used throughout.
Yes

b. Intro & background to show context.
Need to improve according to suggested comments.

c. Literature well referenced & relevant.
Yes

d. Structure conforms to PeerJ standards, discipline norm, or improved for clarity.
Yes

e. Figures are relevant, high quality, well labelled & described.
Yes

Experimental design

a. Original primary research within Scope of the journal.
Yes

b. Research question well defined, relevant & meaningful.
yes

c. It is stated how the research fills an identified knowledge gap.
yes

d. Rigorous investigation performed to a high technical & ethical standard.
yes

e. Methods described with sufficient detail & information to replicate.
Need to improve according to suggested comments.

Validity of the findings

a. Impact and novelty not assessed.
No

b. Meaningful replication encouraged where rationale & benefit to literature is clearly stated.
Yes

c. All underlying data have been provided; they are robust, statistically sound, & controlled.
Yes

Additional comments

To understand the potential role of PTPL1 in DLBCL, the author used DLBCL cell lines. The study shows PTPL1 may be a tumor suppressor gene in DLBCL progression. And also, role of AS2S2 in demethylation of PTPL1. The study is technically proficient and thorough. I have only a few comments and questions for the authors to consider.

1. Your introduction needs more detail. I suggest improving the description in lines 37-38 to provide more information about DLBCL patients. For example, does gender matter in this pathogenesis? What is the survival rate of these patients?

2. In line 46 removes one dot after reference.

3. In line 52-53 add reference at the end of sentence (In DLBCL and follicular lymphoma………).

4. In line 60-62 explain a bit more about the use of AS2S2 as an antitumor medicine and it was used for which tumor treatment and how much it is effective?

5. In line 89, maintained about western blot data but there is no western blot data or result in this manuscript. Therefore, I suggest adding western blot data.

6. In line 91 should mention which plate (96 well plate or 24 well plate) was used for CCK8 assay?

7. In line 97-100 it is not clear which group (NC or experimental) was used for DNA extraction and methylation?

8. In line 112-113 should mention which concentration of AS2S2 was used and this concentration was prepared in which solvent? And also about AS2S2, is this toxic or not?

9. In line 129-133 which group (NC or experimental) was used?

10. In line 150-151 add reference at the end of sentence.

11. In figure2 and 3, should mention the size of the marker that is missing in the manuscript.

·

Basic reporting

Insufficient amount of introduction/review of the literature regarding the background studies about the heterogeneous nature of pathogenesis in DLBCL to establish what type of etiology is in question.

Experimental design

Materials and Methods:
Line 77, 81, PDF annotated: Please be specific here, Reverse transcription PCR (RT-PCR) is being used for making cDNA itself, after that basically, you have run a regular DNA amplification PCR followed by DNA agarose gel electrophoresis (1.5%).
Note: As followed by DNA agarose gel electrophoresis (So this is also not a Realtime Quantitative PCR (qRT-PCR), generally popularly known as RT-PCR.

Results:
Line 128, PDF annotated: You have compared the methylation level of the two DLBCL cell lines with two different primers for the methylated and unmethylated states. Have you used/evaluate/run this experiment and have data for the normal (non-cancerous) cell line or tissue for comparison of the methylation status variation between normal vs disease conditions? That is absolutely necessary to know the methylation pattern variation due to the disease state in comparison to the methylation pattern in a healthy state.

Validity of the findings

1. Biological system doesn't work in isolation, that affects only the particular gene that you want to target, which leads to disturbance of the methylation pattern, leads to demethylation of almost any gene and its promotor in the genome, and that disturbed methylation pattern would results dysregulation of the crucial genes in the biological system, for example, oncogenes demethylation would lead to their upregulation that leads to the development of the high incidence of malignancies, unless you are making some arrangement to deliver particular drug chemical to specific target gene using CRISPR, etc. as tools that specifically deliver chemical to that DNA promoter sequence that would cause de-methylation of that particular gene promoter specifically.

2. The treatment with the As2S2, decreases the expression of the main enzymes DNMT1 and DNMT3a/DNMT3b principally responsible for DNA methylation that helps in genome regulation. If these main methylation enzymes would not work effectively, that would result in a disturbance of the normal methylation pattern, for example, the demethylation effect of As2S2 on oncogenes would result in a higher incidence of cancer due to over-expression of these oncogenes due to its demethylation effect.

---

## Round 0.2 · Minor Revisions

The manuscript is recommended for acceptance with minor revisions. The revisions include adding a relevant figure to accompany the mention of gel electrophoresis (lines 89-91), providing a complete method for western blotting in paragraphs 92-99, ensuring uniform formatting for cell concentration (line 114), and maintaining consistency in formatting the term "AS2O3" throughout the manuscript.

·

Basic reporting

accepted

Experimental design

In lines 89-91, mentioned that amplified products were subjected to gel electrophoresis. However, the figure was not included in the manuscript. Either a relevant figure needs to add or this line needs to remove.

In paragraphs 92-99 Need to add a complete method for western blotting.

In line 114, cell (1×105/mL) should be uniform (such as 1×105/mL) in manuscript.

Validity of the findings

accepted

Additional comments

AS2O3 format needs to be uniform in the manuscript.

---

## Round 0.3 · Minor Revisions

Your manuscript is recommended for acceptance once you address the following minor revisions. Please add a relevant figure for gel electrophoresis, provide a complete method for western blotting, ensure uniform cell concentration formatting (line 114), and maintain consistency in formatting "AS2O3" throughout.

---

## Round 0.4 · Major Revisions

The authors are strongly encouraged to revise the manuscript to address the aforementioned concerns comprehensively. The clarity of the biological rationale, presentation of accurate data, additional experimental work, and data availability are critical factors that will determine the suitability of the manuscript for publication.

In particular, the authors need to address the following issues raised by the section editors:

"How does this data advance our understanding of the controversy regarding the role of this gene (alluded to in 194-200, but which needs careful and thorough expansion in a revision to explain how this work extends or contributes to this issue)?

The authors first and foremost need to provide a clear biological rationale for this work and a clear statement on how this contributes to the discussion. Presently they do not. Any decision on the publication needs to be predicated on this first.

Concerns: The data in figure 1 panel B suggest that shRNA-1 significantly decreases protein levels compared to the NC control yet the extent of this change is very small and I doubt could be accurately measured by immunoblotting. I wish to see the primary data (i.e., all blots of all biological replicates) for all experiments, together with the quantification data. It is essential that this is made available for all readers to judge the veracity of the proposed effects (the previous response given by the authors of not having the blots because of the lack of need to do so was not satisfactory or clearly explained).

We have similar concerns of the proliferation data in Figure 2 – the panels look very similar. Raw data and quantification data are therefore essential for this. How do the authors judge this terribly small effect on proliferation to be biologically relevant? The authors fail to ‘close the loop’ by measuring proliferation rates in response to arsenic disulfide under conditions they could predict would have an effect via demethylation of this gene. This is essential further work. Details of protein loads, conditions for immunoblotting etc need to be provided.

All data (from all biological and technical replicates) must be provided."

---

## Round 0.5 · Minor Revisions

The manuscript has undergone a comprehensive review by two reviewers. Reviewer 1 expresses satisfaction with the authors' responsiveness to their suggestions, noting significant improvements in the paper's overall presentation, experimental design, and validation of findings. Reviewer 1 recommends the paper for acceptance. However, Reviewer 3 highlights the need for improvement in English language usage and provides detailed comments on specific sections, urging the authors to consider professional editing services. Moreover, concerns are raised regarding the potential global effects of As2S2 on regulatory mechanisms and pathways in different cells, leading to unintended consequences such as an increased incidence of cancer. The reviewer emphasizes the importance of addressing the fundamental question regarding the clinical application of As2S2. It is suggested that the authors carefully revise the abstract, clarify statements in the introduction, add headings in the Materials and Methods section, and provide further discussion on the global effects of As2S2 to enhance the manuscript's clarity and scientific rigor. Overall, the manuscript shows promise, but addressing the specific points raised by Reviewer 3 will strengthen its overall quality and impact.

Reviewer 1 ·

Basic reporting

The authors incorporated all my suggestions, and the paper looks in better shape now.

Experimental design

Now it improved.

Validity of the findings

The authors validated the finding using various methods.

Additional comments

I recommend it to accept it.

·

Basic reporting

Professional English: The English language used can be improved by getting some help from some professional services who can thoroughly revise the manuscript for a more professional English version. Some comments were added in an annotated PDF uploaded as a part of the review.

Abstract, Line 16: Be very specific and to the point relevant to the study, avoid general statements: In tumor suppressor genes, hypermethylation is important in the development of cancer. Consider changing, if agree too: "Promoter hypermethylation of the tumor suppressor gene is one of the well-studied causes of cancer development. The drugs that reverse the process by driving demethylation could be a candidate for anticancer therapy.

Line 18: Consider changing the statement: Should write "PTPL Methylation" NOT demethylation", before the experiment and results, the statement that the As2S2 is to cause demethylation seems biased and pre-assumed results. As2S2 causes demethylation, proved by experiments, should be revealed in the results, not in the background.

Line 33: See annotated pdf

Experimental design

Materials and Methods:
Line 156 and 160: Add two different headings to explain these two designed experiments, First, PTLP1 gene methylation status after As2S treatment of different doses in different cell lines. Second, Evaluation of mRNA expression level of different DNMTs on As2S2 treatment with different doses and cell lines. In its present form, It is a bit confusing, the way these two different designed experiments are explained under a single heading, lacks information on how these two different experiments were run and gives results and information to support two different biological processes in the Materials and Methods section (156 to 160) (Concerns explained in details in annotated PDF).
However, results of the these two different experiments are clearly explained in the results section (Lines 181 and 187, and in figure-4 and 5).

Validity of the findings

The present study has provided enough experimental validation for their findings and gives insight into potential clinical applications. The planned research has some limitations as also commented on during the previous revision, related to the targeted gene specificity, so the clinical application is not so straightforward.
So how the authors will answer the very fundamental question regarding the clinical application of As2S2? It targets the DNMTs enzyme family responsible for the Methylation and Demethylation of the genome, a regulatory mechanism to regulate gene expression. The un-targeted global effect of As2S2 will disturb the different normal regulatory mechanisms and pathways in the different cells e.g. The treatment with the As2S2, decreases the expression of the enzymes DNMT1 and DNMT3a/DNMT3b, like, the demethylation effect of As2S2 on oncogenes could result in a higher incidence of cancer due to over-expression of the oncogenes due to its demethylation effect of As2S2.

However, the during previous revision, the authors acknowledged the limitations of the present study and discussed the above limitations at the end of the discussion section.

---

## Round 0.6 · Minor Revisions

The manuscript has improved in response to previous feedback. Findings have been effectively validated with relevant experimental support, strengthening the manuscript's overall quality. Minor revisions, such as ensuring consistency in changes and clarifying ambiguous statements, are recommended before publication.

Note that the Section Editors commented that:

> The authors have improved the rationale and discussion sections well, and to some extent alleviated our concerns.

> However, we note that all of the immunoblots are NOT shown. We strongly urge the authors to upload ALL blots of ALL biological replicates for ALL figures. We note that this issue has not been addressed in their rebuttal, as the supplemental data shows only the blot used in the figure.

> We deem this essential, especially given the small changes in expression levels.

This is also in line with PeerJ policy so please ensure you comply.

·

Basic reporting

Manuscripts have improved considerably as per suggestions.

Experimental design

Changes are added as per the suggestion provided in the previous review. The material and methods section is revised, and now supports and matches with the results presented, largely contributing to improved experimental design. In the discussion and conclusion section, the authors have clearly explained the scope and limitations of the study. As the prospects, the authors also show a great interest in taking the study to a further level with increased scope and a higher level of design.

Validity of the findings

The authors have validated their findings with relevant experimental support within the scope and limitations of the study.

---

## Round 0.7 · accepted · Accept

The article is now in good shape and all the required figures are submitted.